

# A review on the relationship between Arachidonic acid 15-Lipoxygenase (ALOX15) and diabetes mellitus

Kaiying He[1,2,*], Xiaochun Zhou[2,*], Hongxuan Du[1,2], Jing Zhao[1,2], Rongrong Deng[1,2] and Jianqin Wang[2]

[1] Lanzhou University, Lanzhou, Gansu, China
[2] Lanzhou University Second Hospital, Lanzhou University, LanZhou, Gansu, China
[*] These authors contributed equally to this work.

## ABSTRACT

Arachidonic acid 15-lipoxygenase (ALOX15), as one of the lipoxygenase family, is mainly responsible for catalyzing the oxidation of various fatty acids to produce a variety of lipid components, contributing to the pathophysiological processes of various immune and inflammatory diseases. Studies have shown that ALOX15 and its related products are widely distributed in human tissues and related to multiple diseases such as liver, cardiovascular, cerebrovascular diseases, diabetes mellitus and other diseases. Diabetes mellitus (DM), the disease studied in this article, is a metabolic disease characterized by a chronic increase in blood glucose levels, which is significantly related to inflammation, oxidative stress, ferroptosis and other mechanisms, and it has a high incidence in the population, accompanied by a variety of complications. Figuring out how ALOX15 is involved in DM is critical to understanding its role in diseases. Therefore, ALOX15 inhibitors or combination therapy containing inhibitors may deliver a novel research direction for the treatment of DM and its complications. This article aims to review the biological effect and the possible function of ALOX15 in the pathogenesis of DM.

## INTRODUCTION

Diabetes mellitus (DM) is prevalent worldwide, and its incidence is increasing year-by-year. According to the prediction from the International Diabetes Federation, the number of people with DM will reach 700 million by 2045, with the majority of new cases occurring in developing countries such as China and Africa (*Teo et al., 2021*). Various studies have found that inflammation, oxidative stress and apoptosis participate in the development and progression of insulin resistance (IR) and DM. Firstly, the expression of critical pro-inflammatory factors such as interleukin-1β(IL-1β), IL-1, IL-6 and tumor necrosis factor-α (TNF-α) is increased in patients with DM (*Turkmen, 2017*). Meanwhile, these pro-inflammatory mediators are interdependent in inducing tissue-specific inflammation, which may be related to the pathogenesis of IR and DM. In addition, the glycolipid toxicity

Corresponding author
Jianqin Wang,
ery_wangjqery@lzu.edu.cn

of DM can lead to elevated levels of oxidative stress, resulting in increased production of the aforementioned pro-inflammatory factors (*Rendra et al., 2019*). Moreover, hyperglycemia can inhibit the release of insulin and increase the apoptosis of pancreatic β cells. The increase of apoptosis leads to a decrease in the number of cells, ultimately leading to insufficient insulin secretion and the occurrence and development of DM (*Biarnés et al., 2002*). Therefore, it is urgent to search for targets that play essential roles in inflammation, oxidative stress and apoptosis of DM. Studies have shown that the relative protein molecular weight of ALOX15 is about 75kbp, and the isoforms of lipoxygenase are widely distributed in plants, mammals, lower marine organisms and some microorganisms (*Kühn & Borchert, 2002*). In animals, arachidonic acid 12-lipoxygenase (ALOX12) was first identified in human platelets in 1974 (*Hamberg & Samuelsson, 1974*) and named platelet-type 12-LOX. Then, in 1975, *Schewe et al. (1975)* identified a different LOX isoenzyme in the lysate of immature red blood cells that oxidizes membrane lipids and was named reticulocyte type 15-LOX (rabbit Alox15 is the rabbit ortholog of human ALOX15) (*Sigal et al., 1990*). ALOX15 is not expressed in normal erythrocytes but is found in the immature red cell precursors, the reticulocytes. Interestingly, this enzyme is almost undetectable in young reticulocytes, but during *in vitro* red blood cell maturation, the expression of ALOX15 anti-parallels the maturational decline of reticulocyte respiration (*Höhne et al., 1983*; *Rapoport et al., 1979*), suggesting that ALOX15 is associated with the maturation breakdown of mitochondria in the later stages of erythropoiesis (*Schewe et al., 1977*). According to studies, ALOX15 and its related products are expressed at higher levels in many pathological human tissues and organs, which in turn cause inflammation, oxidative stress, and ferroptosis, all of which are closely related to the onset of DM and its complications (*Singh & Rao, 2019*). For ALOX15, an important enzyme discovered many years ago, it was necessary to write a review of its association with DM and DM-related complications.

## GENES GENUS NOMENCLATURE

As for gene genus naming, when it is necessary to identify the origin species of homologous genes with the same genetic symbol, the different species coding based on letters established by SWISS-PROT can be applied. In this review, we use normal letters for genes, small letters for mice and capital letters for human genes.

## SURVEY METHODOLOGY

Publications from 1 January 1985 to 1 January 2023 were retrieved from the Web Of Science, Cochrane Library, PubMed, EMBASE and MEDLINE databases without any language restrictions. We used a mix of MeSH and keywords. With the following terms: (Diabetes Mellitus) OR (Diabetes) OR (Blood glucose, High) OR (High Blood glucose) OR (DM) AND (AA) OR (Arachidonic acid) AND (ALOX15) OR (12/15-LOX) OR (Arachidonate acid 12/15-Lipoxygenase) OR (12/15-lipoxygenase) OR (12-LOX) OR (15-LOX). The final reference list was generated based on relevance and originality concerning the topics covered in this review.

## Arachidonic acid metabolism and diabetes mellitus

Arachidonic acid (AA) is an omega-6 polyunsaturated fatty acid (PUFA) found primarily in the form of phospholipids in cell membranes. When the cell is under stress, AA is released from phospholipids by phospholipase A2 (PLA2), or diacylglycerol (DAG) can be converted to AA *via* DAG lipase activity (*Doherty & Walsh, 1996*; *Sperling et al., 1993*; *Van Dorp, 1975*). The decrease of AA concentration in serum is an early event of IR, and the AA concentration in serum of patients with type 2 diabetes mellitus (T2DM) is significantly decreased. In vitro studies have shown that AA can promote the utilization of glucose by muscle cells, promote the uptake of glucose by adipocytes, inhibit the synthesis of resistin by adipocytes and so on (*Haugen et al., 2005*; *Nugent et al., 2001*; *Rosenthal, Hwang & Song, 2001*; *Steppan et al., 2001*). In addition, AA can synthesize prostaglandins under the action of a series of enzymes after entering the cell. It has been reported that PGE2 and PGE1 (prostaglandin E1/2, the main metabolite of COX pathway) can enhance the insulin sensitivity of rat flounder muscle cells and further affect the metabolism of Zn and enhance the insulin sensitivity of cells (*Ezaki, 1989*; *Leighton et al., 1985*). *Dixon et al. (2004)* reported *in vitro* studies that AA can improve the ability of pancreatic β-cell to secrete insulin. A cohort study of T2DM patients found that circulating AA was negatively associated with urinary albumin excretion (UAE) and macroalbuminuria (*Okamura et al., 2021*). AA has been reported to enhance hypoxia-induced vascular endothelial growth factor (VEGF) expression through Notch-1, Wnt-1 and Hif-1α pathways (*Okamura et al., 2021*). Previous studies have shown that VEGF can improve diabetic kidney disease (DKD), normalize glomerular hyperpermeability and restore endosylcalyces in DKD (*Falkevall et al., 2017*; *Flyvbjerg et al., 2002*; *Oltean et al., 2015*). Studies have shown that in men, AA is negatively correlated with the risk of DM (*Wu et al., 2017*). In addition, a lower incidence of type 1 diabetes mellitus (T1DM) has been reported in people who breastfeed for more than three months (human breast milk is rich in various PUFAs, especially AA) (*Grzywa & Sobel, 1995*).

At present, at least three metabolic pathways (COX pathway, CYP450 pathway and LOX pathway) are known to participate in the metabolism of AA. The relationship between the first two metabolic pathways and DM has been studied extensively. The following part summarizes the latest progress of ALOX15 (one of the research hotspots in the LOX pathway) between DM and its complications.

## Introduction of ALOX15 in the lipoxygenase family
### ALOX15 genotype and structural characteristics

Lipoxygenase (LOX) is a kind of non-heme iron-containing fatty acid dioxygenase, which can catalyze PUFAs with cis, cis-1 and 4-pentadiene structures into specific hydroperoxy derivatives. The human genome involves six functional LOX genes that encode six different LOX subtypes (ALOX5, ALOX12, ALOX12B, ALOX15, ALOX15B, ALOXE3). Interestingly, there are two types of ALOX15 homologous genes in mammals. Highly developed primates (humans, chimpanzees, orangutans) express an arachidonic acid 15-lipoxygenating ALOX15, whereas lower primates and other mammals express an arachidonic acid 12-lipoxygenating enzyme. Mouse Alox12, Alox12b, Aloxe3, and Alox5

have a high degree of amino acid conservation with their human congeners and exhibit similar enzymatic properties, while a similar degree of amino acid conservation was found for the Alox15 orthologues of rats, pigs and cattle (*Kuhn et al., 2018*; *Kuhn, Banthiya & van Leyen, 2015*). However, this is not the case with mouse Alox15 and Alox15b. In fact, mouse Alox15 is an arachidonic acid 12-lipoxygenating enzyme that mainly converts AA to 12S-hydroxy-peroxy-eicosatetraenoic acid (12S-H(p)ETE) (*Kühn et al., 1993*). In contrast, the human ortholog exhibits an arachidonic acid 15-lipoxygenating activity, primarily converting AA to hydroperox eicosatetraenoic acid (*Chen et al., 1994*).

Human ALOX15 widely exists in eosinophils, macrophages, bronchial epithelial cells and skin, and can convert linoleic acid (LA), AA and other PUFAs into active lipid metabolites, thereby affecting cell structure, metabolism and signal transduction. Human ALOX15 is initially present in the late stage of reticular cell maturation and immature red blood cells and is highly expressed in eosinophils and bronchoalveolar epithelial cells (*Nadel et al., 1991*). Human immature red blood cells can express ALOX15, but in different species (human, rabbit, mouse, rat), mature red blood cells do not express ALOX15. The expression of this enzyme is upregulated in immature red blood cells during experimental and natural anemia (*Kroschwald et al., 1989*; *Ludwig et al., 1988*; *Schewe et al., 1990*). In addition, human and mouse peripheral blood mononuclear cells do not express ALOX15 in circulation, but IL-4 and IL-13 induce mRNA and protein expression of ALOX15 in human mononuclear cells and mouse macrophages *in vitro* (*Brinckmann et al., 1996*; *Conrad et al., 1992*; *Heydeck et al., 1998*). In human umbilical vein endothelial cells, IL-4 induces ALOX15 mRNA expression but does not induce active enzyme expression (*Lee et al., 2001*). Interestingly, because 10%–40% of cells do not express ALOX15, IL-4 did not induce ALOX15 expression in all peripheral monocytes (*Kühn & O'Donnell, 2006*). The reason for this heterogeneity is unclear, but it may be related to the maturation stage of cells and their metabolic status (*Tsao et al., 2014*).

### Human ALOX15 substrates and metabolites

In DM, AA acts as a strong inducer of insulin secretion, but the contribution of its metabolites to IR depends on the cells and tissues involved (*Luo & Wang, 2011*). AA can be converted to leukotrienes (LTs) and lipotoxins (LXs) through the LOX pathway (*Calder, 2015*; *Rae, Davidson & Smith, 1982*; *Yates, Calder & Ed Rainger, 2014*), thus widely participating in a variety of physiological and pathological processes (*Anonymous, 1981*; *Kopp et al., 2019*). AA, as one of the primary substrates of ALOX15 *in vivo*, is mainly oxidized to 15-H(p)ETE in humans (*Kühn et al., 1993*; *Sigal et al., 1990*), while rats (*Pekárová et al., 2015*; *Watanabe & Haeggström, 1993*) or mice (*Freire-Moar et al., 1995*; *Sun & Funk, 1996*) congeners mainly produced 12-H(p)ETE. These data indicate that the structure of the catalytic center of ALOX15 is slightly different between rodents and humans. These above differences suggest that experimental data on LOX metabolism should be handled with caution if transferred from one species to another (*Funk et al., 2002*; *Kuhn, 2004*). Next, under the action of cellular glutathione peroxidase (GSH-Px), 15(S)-H(p)ETE and 12(S)-H(p)ETE were further reduced to 15-hydroxy eicosapenoic acid (15(S)-HETE) and 12-hydroxy eicosapenoic acid (12(S)-HETE), respectively. 12(S)-H(p)ETE can be

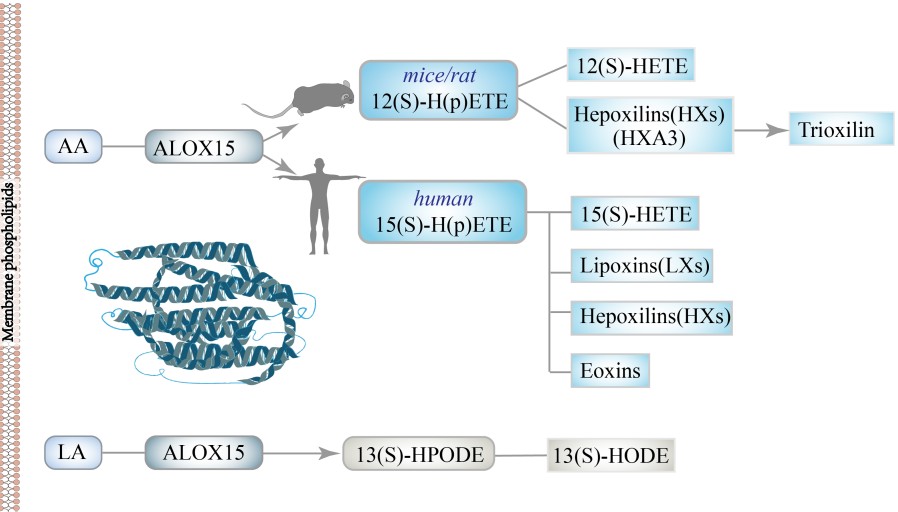

**Figure 1** **The main substrates and metabolites of ALOX15.**

metabolized to hepoxilin, which is further metabolized to trioxilin, the corresponding tri-hydroxyl metabolites. These metabolites have been shown to induce insulin secretion in animal models of DM (*Funk, 1996*). Meanwhile,15(S)-H(p)ETE and 15(S)-HETE could be metabolized into different bioactive lipids, for instance, lipoxins, hepoxilins and eoxins (*Kühn & O'Donnell, 2006*). The ratio of 12(S)-HETE to 15(S)-HETE catalyzed by Alox15 varies among species; the ratio is about 3:1 in mice, 6:1 in rat brain tissue, and 11:1 in bovine bronchus (*Funk et al., 2002*). In addition, using LA as substrate, ALOX15 can be dominantly metabolized into 13(S)-hydroperoxyoctadecenoic acid (13(S)-HPODE), and then the metabolites can be further metabolized into 13(S)-HODE (*Kutzner et al., 2017*). 13(S)-HODE was found to be involved in neuronal activation, lipid metabolism and monocyte maturation through peroxisome proliferators-activated receptors (PPAR), transient receptor potential cationic channel subfamily V member 1 (TRPV1) and G protein-coupled receptor 132 (GRP132). The main substrates and metabolites of ALOX15 are shown in Fig. 1.

## Biological effects of ALOX15
### The dual effect of ALOX15 on inflammation
Inflammation is one of the leading causes of IR, and in DM, the expression of many pro-inflammatory factors is increased, such as IL-1β, IL-6, IL-8, IL-12, TNF-a and so on. Since the immune regulatory system of the human body has an obvious regulatory mechanism, the enhancement of the pro-inflammatory response will negatively stimulate the activation of the anti-inflammatory response, which also means that the anti-inflammatory factor will be increased, for example, IL-4, IL-10, IL-11, IL-13, to inhibit the tissue damage caused by the excessive inflammatory response (*Burhans et al., 2018*).

Lipoxins (LXs), one of the downstream products of the oxidation of a series of essential PUFAs by ALOX15, have significant anti-inflammatory effects when their synthetic levels

are elevated (*Chan & Moore, 2010*). *Chinthamani et al. (2012)* found that LXs can play an anti-inflammatory role by inhibiting the adhesion of immune cells to vascular endothelial cells and up-regulating the expression of vascular cell adhesion molecule-1 (VCAM-1). Moreover, LXs can protect the expression of IL-4, IL-10, IL-13 through the MAPK pathway (ERK, P38, JNK) and PIPP pathway, in order to play a certain anti-inflammatory role. Current studies have shown that when AA is deficient, the formation of LXA4, one of the metabolites of LXs, is reduced, and the production of PGE is increased, ultimately leading to pancreatic β cell dysfunction and the occurrence of DM (*Gundala, Naidu & Das, 2018*). It also found that plasma phospholipid content of AA(*Das, 1995*) and circulating level of LXA4 (*Kaviarasan et al., 2015*) are lower in patients with T2DM. In addition, in streptozotocin (STZ)-induced animal models of T2DM, oral intake of AA can inhibit the IL-6 and TNF-α production, and STZ-induced inhibition of LXA4 production returned to normal, thereby completely preventing hyperglycemia and improving insulin sensitivity (*Gundala, Naidu & Das, 2018*). Sufficient AA can promote the formation of LXA4, a potent anti-inflammatory compound, which can antagonize the pro-inflammatory effect of leukotrienes (*Tan et al., 2022*). At the same time, AA can inhibit the expression of NF-kB in the pancreas and adipose tissue, and enhance the expression levels of ALOX5 and ALOX15, which may be the reason for its anti-diabetic effect. In T2DM, AA brings plasma TNF-α and IL-6 levels back to normal levels, which may be responsible for the return of insulin sensitivity to normal (*Gundala, Naidu & Das, 2018*). In conclusion, AA and LXA4 have a protective effect on DM (*Das, 2013*; *Gundala, Naidu & Das, 2017a*; *Gundala, Naidu & Das, 2017b*; *Suresh & Das, 2001*). Of note, other unsaturated fatty acids: gamma-linolenic acid (GLA, 18:3n-6), eicosapentaenoic acid (EPA, 20:5n-3), and docosahexaenoic acid (DHA, 22: n3) have also shown cytoprotective effects *in vitro* against pancreatic beta cytotoxicity in alloxan and STZ induced T1DM and T2DM in experimental animals, although their beneficial effects are much weaker than AA (*Krishna Mohan & Das, 2001*; *Suresh & Das, 2001*; *Suresh & Das, 2003a*; *Suresh & Das, 2003b*). We hypothesized that EPA and DHA may replace AA in the lipid pool of the cell membrane, thus promoting the production of LXA4. However, recent studies have challenged the biosynthesis and biological relevance of LXs. The study showed that the putative deletion of bio-synthetase did not provide evidence consistent with the role of LXs in resolving inflammation (*Schebb et al., 2022*). Therefore, the evidence for the role of LXs in promoting inflammatory regression through specific receptors is still controversial and incomplete, and needs further study. At the same time, 15-HETE, a metabolite of ALOX15, acts as an endogenous ligand of PPAR-γ. Activated PPAR-γ takes part in inflammatory response by deterring the release of TNF-a, iNOS through PI3K/AKT/NFkB and PI3K/p38MAPK/NFkB pathways (*Paintlia et al., 2006*). To sum up, as one of the endogenous metabolites of ALOX15, LXs is a strong anti-inflammatory factor, which is known as the "brake signal" or "stop signal" of inflammatory response.

In addition, studies have shown that ALOX15 also has pro-inflammatory effects. Alox15 catalyzes LA to produce 13-HPODE and induces MCP-1 production in blood vessels through activation of NF-kB, thus promoting inflammation (*Dwarakanath et al., 2004*). *Lindley et al. (2010)* indicated that the augmented expression level of Alox15 aggravates airway allergic inflammation, and the inflammatory mediators produced

after the inflammatory reaction upregulate the level of Alox15 *in vivo*. These above results confirmed that Alox15 promoted the inflammatory response. Moreover, lipopolysaccharide (LPS) induced impaired IL-12 synthesis in Alox15 knockout mice, indicating that it is participated in the secretion of inflammatory factors (*Middleton et al., 2009*). Recent studies found that Alox15 metabolites can stimulate the expressions of IL-6 and TNF-a in a dose-dependent manner. In Plox-86 cells capable of constant expression of Alox15, the amount of Alox15 metabolites was approximately 3.6-fold higher and the expression of cytokines was 2–7-fold higher than in normal cells. 12S-HETE promotes the production of inflammatory factors mainly by activating mitogen-activated protein kinases (P38, MAPK, JUK), protein kinase C (PKC) as well as other kinases. After blocking the activity of these kinases, the expression of IL-6 and TNF-a were dramatically diminished (*Wen et al., 2007*). The biological activity of ALOX15 itself may promote inflammation (*Uderhardt & Krönke, 2012*). These researches indicate that ALOX15 can directly stimulate cytokine expression and induce inflammatory cascades. In conclusion, ALOX15 has the dual effect of promoting or inhibiting inflammation. While LOX-mediated AA metabolites (such as 5-HETE and leukotriene B4 (LTB4) from ALOX5-mediated metabolism) help initiate acute inflammation (*Funk, 2001*), other products of LOX-mediated PUFAs metabolism (lipotoxins (from AA)), resolvins (from DHA and EPA, protective proteins (DHA) and fatty acids (DHA)) are essential for the active process of inflammation resolution, the failure of which leads to the development of chronic inflammation (*Serhan, 2014*).

### The effect of ALOX15 on oxidative stress

Except for its significant role in inflammation, ALOX15 is also related to oxidative stress in the human body. During cell metabolism, lipid oxidase ALOX15 interacts with GSH-Px to regulate cell redox state and apoptosis pathway. GSH-Px is a main peroxide-reducing enzyme which broadly exists in the human body. It can catalyze the transformation of GSH into GSSG and reduce toxic peroxide into non-toxic hydroxyl compounds, so as to ensure the function and structure of the cell membrane are not disturbed and damaged by oxides. In oxidative stress, oxygen molecules generate superoxide anions under the metabolic action of oxidase and mitochondria. Superoxide anions act on cell membranes after transformation to promote the transformation of $Fe^{2+}$ into $Fe^{3+}$ and activate ALOX15, which catalyzes the formation of 12(S)-H(p)ETE from AA on membrane phospholipids, and in the presence of glutathione peroxidase-4 (GPX4), ALOX15 can be isomerized to form hepoxilins or 12(S)-HETE (*Schnurr, Borchert & Kuhn, 1999*). GPX4 can regulate ALOX15 to prevent the conversion of 12S-H(p)ETE to 12S-HETE and isomerize into hepoxilins. In turn, ALOX15 also regulates GPX4 and promotes the redox process of GSH-dependent membranes, and its decreased activity may have a complementary effect on GPX4. These two enzymes play antagonistic roles in lipid peroxidation metabolism (*Kühn & Borchert, 2002*; *Schnurr, Borchert & Kuhn, 1999*), and it has been shown that Alox15$^{-/-}$ cells consume GSH through L-Buthionine sulfoximine (BSO) to resist GPX4 inhibition. Therefore, their balanced expression is important for intracellular redox homeostasis. ALOX15 homologs oxidize complex lipids carrying PUFAs to corresponding hydroperoxides, which may initiate secondary oxidation reactions (*Ivanov, Kuhn & Heydeck, 2015*; *Singh & Rao, 2019*). GPX4

homologs reduce this complex hydroxyl lipid to a core-responsive hydroxyl derivative at the expense of reduced glutathione, and as a result, the enzyme reduces the cell oxidation potential (*Brigelius-Flohé & Maiorino, 2013*; *Schnurr et al., 1996*). The apoptotic pathway induced by GPX4 inactivation can be blocked by the application of ALOX15 inhibitors. GSH is a synergistic substrate of GPX4, and ALOX15-deficient cells are remarkably immune to oxidative damage caused by GSH deficiency (*Seiler et al., 2008*). In addition, the expression of ALOX15 augmented the secretion of reactive oxygen species (ROS) and free radicals, which increased NADPH oxidase activity, promoted NADPH oxidation, decreased cytochrome C activity, and promoted the oxidative stress process (*Othman et al., 2013*).

### The effect of ALOX15 on ferroptosis

Ferroptosis belongs to programmed iron-dependent cell death, a new cell death mode proposed by *Dixon et al. (2012)*. The main pathophysiological manifestations of ferroptosis include increased iron ions, enhanced lipid peroxidation, smaller mitochondria and increased membrane density, including oxidative stress. The present study showed that ferroptosis is involved in the study of the pathogenesis of various diseases, including ischemic tissue damage (such as brain damage (*Gou et al., 2020*), ischemic heart disease (*Del Re et al., 2019*)), renal failure (*Wang et al., 2021*) and acute lung injury (*Liu et al., 2020*), neurodegenerative diseases (such as Alzheimer's disease (*Bao et al., 2021*), Parkinson's and Huntington's disease (*Reichert et al., 2020*)), tumor diseases (such as breast cancer (*Li et al., 2020*), colorectal, lung, liver, pancreatic and urologic cancer (*Wei et al., 2021*; *Ye et al., 2021*; *Zhang et al., 2021*; *Zhang et al., 2019*; *Zhao et al., 2021*)), autoimmune encephalomyelitis (*Li et al., 2022*), *etc.* ALOX15 can catalyze the stereotactic oxygenation of PUFAs such as AA and LA, leading to lipid peroxidation and further causing ferroptosis (*Lee et al., 2021*). Studies suggested that silencing ALOX15 significantly reduces ferroptosis induced by erastin or RSL3 (Ras-selective lethal small molecule 3), while overexpression of exogenous ALOX15 increased ferroptosis. At the same time, the immunofluorescence results demonstrated that ALOX15 was always localized in the cell membrane during the process of ferroptosis. These results suggest that LOX-catalyzed hydrogen peroxide production of cell membrane lipids promotes tumor ferroptosis (*Shintoku et al., 2017*). Notably, in a large number of cell model systems, ferroptosis occurred even though ALOX15 was not expressed in these cells, such as human mature red blood cells (*Chen et al., 2022b*). Therefore, ALOX15 is not a necessary participant in ferroptosis but only has a vital role in the ferroptosis of those cell lines that express ALOX15.

In addition, ALOX15 has been reported to be involved in the occurrence of cell ferroptosis in RSL3-induced acute lymphoblastic leukemia (ALL). They found that RSL3, as a GPX4 inhibitor, leads to increased lipid peroxidation, ROS production, and cell death in ALL cells. Meanwhile, LOX inhibitors can protect cells from the above effects of RSL3 stimulation. Therefore, these results indicated that ALOX15 can promote the occurrence of ferroptosis in cells (*Probst et al., 2017*). The esterification and oxidation of PUFAs are vital in promoting ferroptosis, and studies have found that ALOX15 participates in this process, suggesting that it may be a significant regulator in ferroptosis. Studies have shown that phosphatidyl

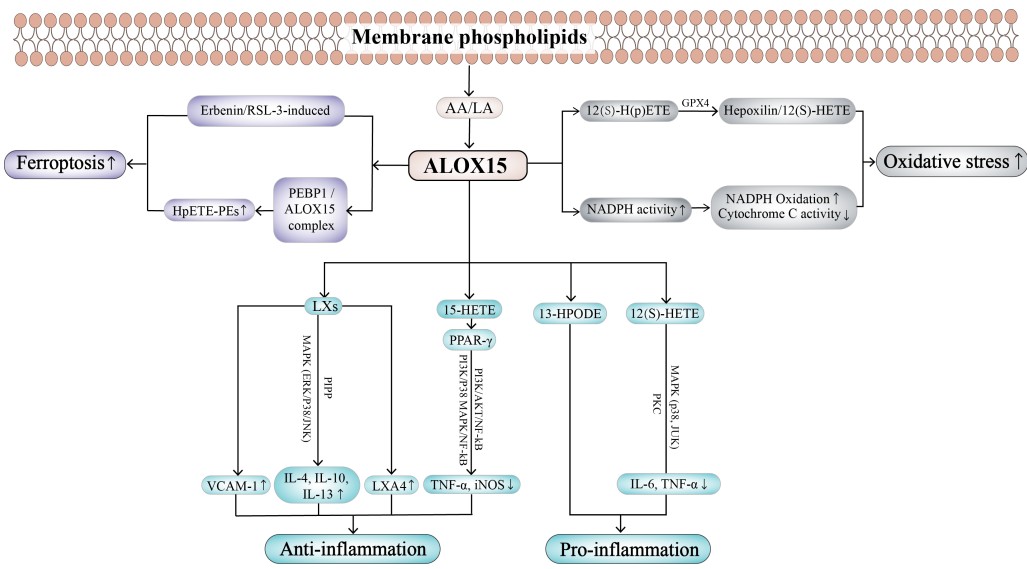

**Figure 2  The biological effects of ALOX15 in different diseases.**

ethanolamine binding protein 1 (PEBP1) binds to ALOX15 to form PEBP1/ALOX15 complex, which can change the specificity of PUFA-PE catalyzed by ALOX15 to form HpETE-PEs from free PUFAs. Due to insufficient or dysfunctional production of GPX4, excess production of HpETE-PEs cannot be eliminated in time, resulting in ferroptosis (*Stoyanovsky et al., 2019*). Thus, the PEBP1/ALOX15 complex is regarded as a main regulator of ferroptosis in airway epithelial cells from asthma patients, renal epithelial cells from renal failure patients, and cortical and hippocampal neurons from brain trauma patients (*Stoyanovsky et al., 2019*). The biological effects of ALOX15 in different diseases are shown in Fig. 2.

## ALOX15 in the pathogenesis of DM and its associated complications
### ALOX15 and islet cell dysfunction

ALOX15 and its metabolites are associated with DM. The main function of islet β cells is to secrete insulin, which can promote the uptake and utilization of glucose, and convert glucose into protein, fat and glycogen, thereby reducing blood glucose (BG) levels (*Leto & Saltiel, 2012*). As early as 1999, *Bleich et al. (1999)* reported that in the established drug-induced DM mice model, C57BL/6J mice suffered islet cell damage similar to T1DM, leading to a significant increase in the incidence of DM in mice. However, in low-dose STZ-induced DM models, Alox15 knockout mice showed reduced damage to islet cells and reduced incidence of DM (*Bosma et al., 2022*). A study in 2017 also confirmed that reducing the Alox15 expression level is an effective treatment for glycemic exacerbation in T1DM (*Hernandez-Perez et al., 2017*). In addition, studies of obese rats with IR have also suggested that the expression level of Alox15 in adipose tissue of obese rats induced by high-fat diet (HFD) is higher than that of normal controls, and the use of Alox15 inhibitors could antagonize IR in obese rats. Similarly, reduced expression of Alox15 significantly

improved the occurrence of T1DM in non-obese diabetic (NOD) mice (*Green-Mitchell et al., 2013*). In conclusion, the expression of ALOX15 can lead to the dysfunction of islet cells, which may play a part in the occurrence of DM in obese patients (*Dobrian et al., 2011*).

It is well known that damage to islet β cells can lead to the occurrence of both T1DM and T2DM (*Cnop et al., 2005*). Studies have found that different concentrations of AA had different effects and gradually showed toxic effects on islet β cells with the increase of its dose (*Keane et al., 2011*). *Nunemaker et al. (2008)* and *Sears et al. (2009)* identified the function of Alox15 in islets by gene knockdown and targeted protein knockdown, which showed that reducing Alox15 expression could prevent islet function impairment as well as IR induced by HFD. Moreover, *Tokuyama et al. (1995)* found that Alox15 expression levels were elevated in diabetic Zucker rats, and β-cells were the preferred cells for Alox15 expression. Meanwhile, it has been shown that the expression of Alox15 coexists with α-cells (secrete glucagon) of rat islets, and the increased expression of Alox15 in α-cells can promote the secretion of glucagon, thereby increasing BG level (*Kawajiri et al., 2000*). Moreover, in human islets, an increase in exogenous 12S-H(p)ETE or 12S-HETE lead to a significant decline in islet activity and insulin production (*Ma et al., 2010*). A recent review by Nadler's group discussed the harmful effects of 12-HETE on the onset of DM (type 1 and type 2) and obesity in Alox15 knockout mice (*Kulkarni et al., 2021*). Moreover, 12S-HETE can increase the apoptosis of islet β cells by increasing mitochondrial oxidative stress (*Nazarewicz et al., 2007*). Although 12S-H(p)ETE is the main product of Alox15 in mice models and not human ALOX15, it plays a vital role in DM. The researchers only used physiological concentrations of 12(S)-HETE and did not test its enantiomer (12(R)-HETE) or the specific role of 15-H(p)ETE/15S-HETE, the main product of human ALOX15 in AA, in β-cell apoptosis.

However, it is noteworthy that although AA can stimulate insulin secretion by β-cells in the pancreas, ALOX15 constrains insulin secretion, possibly because of the reduction of free AA (*Persaud et al., 2007*). It has also recently been suggested that the Alox15 cell-specific deletion in islets improve insulin secretion and protects the body from abnormal BG level associated with HFD. Besides, it has been reported that the Alox15 level in islets is significantly increased in db/db mice models from 10 weeks of age, besides the extent of the growth in Alox15 level is consistent with the extent of the reduction in islet cell number (*Dobrian et al., 2018*). Notably, the NOD mice congenic for a targeted deletion of Alox15 are protected from autoimmune DM (*McDuffie et al., 2008*). These results suggest that Alox15 plays a role in both the pathology and mechanism of DM, which may be related to the effect of Alox15 on the function of islet cells and macrophages (*Green-Mitchell et al., 2013*). Although the underlying mechanism is unknown, the results suggest an interaction between Alox15 expression in adipose tissue and islet inflammation, and inhibition of Alox15 expression in adipose tissue may provide systemic protection against obesity-induced consequences. Moreover, blocking the activity of Alox15 in adipose tissue may constitute a new therapeutic principle for the treatment of T2DM (*Cole et al., 2013*).

Hepoxilins (HXs) are bioactive epoxy hydroxyl products metabolized by AA through the 12S-lipoxygenase pathway. After the dioxygenation of AA with 12S-H(p)ETE, a recently

discovered enzyme, hepoxilin A3 (HXA3) synthetase, readily converts 12S-H(p)ETE into biologically active compounds, 8S/R-hydroxy-11, 12-epoxy-5Z,9E, 14z-trienoic acid (hepoxilin A3), commonly known as HXA3, and the inactive compounds 10S/R-hydroxy-11, 12-epoxy-5Z,8Z, 14z-trienoic acid (HXB3)(*Nigam et al., 2007*). HXs can participate in various physiological processes, such as the release of inflammatory mediators, insulin secretion, calcium regulation, potassium regulation, *etc* (*Newman, Morisseau & Hammock, 2005*). Studies have found that exogenous HXA3 can increase the release of insulin after acting on the Langerhans islets of well-perfused rats (*Pace-Asciak & Martin, 1984*). After injection of the HXA3 isomer (100 μg HXA3/rat), circulating insulin levels were enhanced within 20 min. In an earlier study, researchers found that Langerhans' islets were able to convert 12S-H(p)ETE to HXA3 and HXB3 in addition to 12-HETE, thus demonstrating the activity of HXs synthetase. At a concentration of 2 μM, the glucose produced by HXA3 (10 mM) stimulated the release of insulin almost three times as much as the control group (*Pace-Asciak, 2015*). Subsequently, studies have shown that HXs can release insulin in the body after intra-arterial administration in rodents (*Pace-Asciak et al., 1999*). The effect of HXA3 on insulin release from β cells may be caused by the specific mobilization of calcium as a direct effect on the endoplasmic reticulum (*Dho et al., 1990*). However, whether HXs are formed in Alox15 knockout mice has not been studied, so it is impossible to determine whether they are responsible for additional anti-diabetic effects.

### *ALOX15 and diabetic retinopathy*

Diabetic retinopathy (DR) is a common complication of DM, which can lead to blindness in severe cases (*Cheung, Mitchell & Wong, 2010*). Previous studies in human patients and animal models have demonstrated that ALOX15/Alox15 is highly expressed in the retina (*Al-Shabrawey et al., 2011*). Recent studies have found that Alox15 is involved in vascular hyper-permeability during DR through NADPH oxidation-dependent mechanisms, including inhibition of protein tyrosine phosphatase and activation of the VEGF receptor 2 (VEGF-R2) signaling pathway. At the same time, inhibiting the expression of Alox15 decreased the levels of retinal inflammatory cytokines, the production of ROS, and the expression of phosphorylated VEGF-R2 in DM mice (*Othman et al., 2013*). However, there is no clear cellular origin of the above metabolites, and it is reasonable to assume that they may come from retinal tissues, including retinal vascular endothelial cells, glial cells and pigment epithelial cells, as well as infiltrating inflammatory cells. In addition, *Augustin et al. (1997)* found the expression of the ALOX15 metabolite 15S-HETE was hugely increased in the retinal adventitia of patients with DR in the 1990s. Subsequent studies found that 5S-HETE, the main product of ALOX5, was significantly increased in the vitreous body of DM patients, while the level of 15S-HETE was not significantly changed (*Ibrahim et al., 2015*). In addition, 5S-HETE, 12S-HETE, and 15S-HETE have also been reported to be crucial in diverse phases of DR (*Othman et al., 2013*). Intravitreal injections of 12-HETE have been studied to produce many early DR characteristics, including pro-inflammatory responses and edema. Secondly, 12/15-HETE enhanced several *in vitro* endothelial cell activities related to barrier function and angiogenesis, including reduced resistance, adhesion response to polymorphonuclear neutrophils, migration, and tube

formation (*Graeber et al., 1990*). 15-HETE activates retinal endothelial cells through the NOX system, resulting in increased white blood cell adhesion, high permeability, and retinal neovascularization, a major symptom of DR (*Ibrahim et al., 2015*).

### ALOX15 and diabetic peripheral neuropathy

Diabetic peripheral neuropathy (DPN), a common neuropathy caused by DM, has an incidence of about 30% to 90% and is a leading reason for amputation (*Boulton et al., 2005*). Recent studies have shown that ALOX15 is highly expressed in sciatic nerve, spinal cord and dorsal root ganglion (DRG) neurons of mice as well as human Schwann cells, and its expression is increased in DM and high glucose (*Stavniichuk et al., 2010*), and the expression level of Alox15 in peripheral nerves and DRG of mice fed with HFD is significantly increased (*Obrosova et al., 2007*). Increased expression of Alox15 in the sciatic nerve of DM-induced mice was associated with enhanced oxidative stress and PARP activation, and compared to untreated wild-type mice with DM, these two phenomena were less present in Alox15$^{-/-}$ mice or Alox15-inhibitor-treated DM wild-type mice (*Stavniichuk et al., 2010*). Therefore, it is very plausible that Alox15 overexpression partially explains the enhanced oxidative nitrosation stress and PARP activation of peripheral nerves and DRG neurons in HFD-fed mice. Although LOX overexpression and activation play critical roles in pain functional changes of large and small fibers as well as axonal atrophy of large myelin fibers in DPN, it has little function in epidermal nerve fiber loss, sympathetic plant ganglion dystrophy and neuronal degeneration. These results indicate the functional changes of ALOX15 in DPN and the important role of ALOX15 in demyelination of DPN, providing a theoretical basis for further use of ALOX15 inhibitors or combination therapy containing ALOX15 inhibitors (*Coppey et al., 2021*). It has also been reported that PM5011, a substance extracted from Artemisia annae, can reduce the level of Alox15 from STZ-induced DM mice and improve DPN (*Watcho et al., 2011*). With regard to diabetic cognitive dysfunction, Alox15 can promote inflammation and neuronal apoptosis by activating p38/MAPK and participating in diabetic brain injury. Meanwhile, intervention with Alox15 can improve the above changes (*Chen et al., 2022a*).

### ALOX15 and diabetic kidney disease

Diabetic kidney disease (DKD) is one of the main complications of DM patients. The incidence of DKD in China is also increasing, and it has become the leading reason of end-stage renal disease (ESRD). Studies have shown that lipid mediators and oxidized lipids, especially the LOX metabolic pathway of AA, are involved in the pathological processes of various renal diseases, including DKD. Various reports have indicated that Alox15 is essential in DKD (*Dobrian et al., 2011*). Immunohistochemical (IHC) results showed that Alox15 was located in glomerular mesangial cells, podocytes and microvessels, and its expression level increased with the progression of DKD (*Kang et al., 2001*). The level of ALOX15 was greatly upregulated in glomerular mesangial cells cultured with high glucose (*Xu et al., 2006*). Besides, TGF-β or angiotensin II (AngII) stimulation increased the expression of ALOX15 in glomerular mesangial cells, while the inhibition of ALOX15 levels can reduce the corresponding glomerular mesangial cell hypertrophy and matrix production (*Kim et al., 2003*; *Yiu et al., 2003*). Simultaneously, silencing the expression of

Alox15 by siRNA significantly attenuates renal dysfunction in the mice models of T1DM (*Yuan et al., 2008*).

The research reported that in a mouse model of T1DM kidney disease, the mRNA and protein levels of Alox15 are positively correlated with the expression level of fibronectin, and the urinary excretion rate of 12S-HETE is increased. Researches have suggested that ALOX15 and its metabolite 12S-HETE can stimulate the expression level of COX2 and promote the production of PGE2, which in turn can increase the level of ALOX15 and 12S-HETE. Another study points out that ALOX15 and COX2 work together to aggravate the pathological process of DKD (*Xu et al., 2006*). *In vitro* experiments showed that 12S-HETE could directly stimulate the proliferation of rat mesangial cells, lead to the expression of fibronectin, the migration and proliferation of vascular smooth muscle cells (VSMCs), and regulate the growth of mesangial cells and the production of extracellular matrix. In addition, 12S-HETE stimulated VSMCs in the same way as AngII, and 12S-HETE regulated them in part through P38/MAPK and its target transcription factor cAMP response element-binding protein (CREB)(*Reddy et al., 2002*). Knockdown of Alox15 reduced the growth, matrix production, oxidative stress response, and activation of MAPK and CREB in rat mesangial cells. In addition, in obese nephropathy, ALOX15 also tends to be highly expressed, and the growth in its expression level is related to the activation of P38/MAPK as well as ERKl/2 pathways (*Xu et al., 2005*). Figure 3 shows the role of ALOX15 in the pathogenesis of DM and its related complications.

## Summary of ALOX15 inhibitors in a variety of diseases
### Baicalin

In neurodegenerative disorder diseases, hippocampal neuronal cells (HT22) were treated with a dose of baicalin (the selectively ALOX15 inhibitor), which resulted in inhibition of ERK and decreased ROS production (*Stanciu et al., 2000*). Moreover, studies have indicated that baicalin can significantly reduce the behavioral deficits after embolic stroke in rabbits, suggesting that baicalin or baicalin derivatives can be developed as a new therapy for acute ischemic stroke (AIS) (*Lapchak et al., 2007*). It has been reported that ischemia-reperfusion injury can induce PPAR-γ expression and translocation, whereas baicalin can reverse the above PPAR-γ changes (*Xu et al., 2010*). In prostate cancer, the use of baicalin sensitized tumor cells to radiation, but not normal cells (*Lövey et al., 2013*). Besides, researchers have reported that baicalin may reduce the level of VEGF in human PC-3 cells, thereby mediating the angiogenesis of prostate cancer (*Nie et al., 2006*).

### PD146176

Recent researches indicated that PD146176, another specific inhibitor of ALOX15, can reduce plasma 12-HETE levels, resulting in increased calcium deposition in the aortic arch and vascular calcium content (*Han et al., 2021*). It has also been reported that PD146176 can reduce the epithelial-mesenchymal transition (EMT) in eosinophilic chronic rhinosinusitis (ECRS) with nasal polyps after inhibiting ALOX15 expression (*Yan et al., 2019*). In addition, in kidney disease, compared with wild-type animals, renal inflammation, fibrosis and macrophages infiltration were significantly reduced after UUO

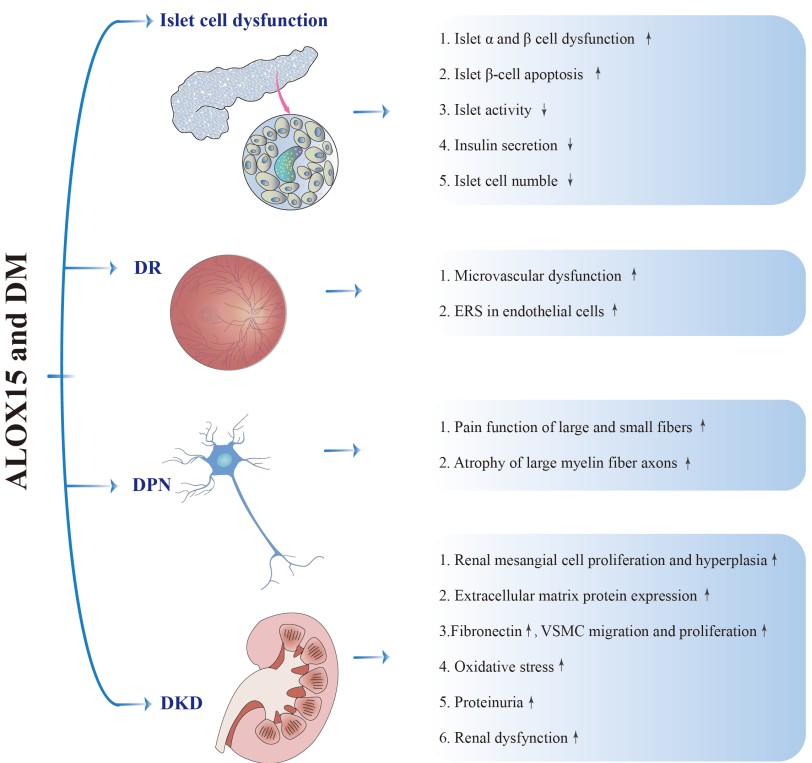

**Figure 3  ALOX15 in the pathogenesis of DM and its associated complications.**

treatment in Alox15 knockout mice. PD146176 also reduced the above changes, and the reduction was similar to that of Alox15 knockout mice (*Montford et al., 2022*).

### BHPP

An ALOX15 inhibitor, N-benzyl-N-hydroxy-5-benzamidine (BHPP), has been shown to reduce 12-HETE in urine and significantly improve DKD over four months of treatment. Administration of BHPP (3 mg/kg/day) reduced 12-HETE/creatinine (cr) in urine by about 30%–50% of DM rats after one week, and at the same time, the excretion rate increased in the corresponding control group. Because urinary 12-HETE/cr excretion was highly correlated with urinary albumin /cr ($r = 0.79$, $P < 10^{-5}$), indicating renal ALOX15 expression is related to proteinuria, and the use of ALOX15 inhibitor can alleviate proteinuria in DKD (*Ma et al., 2005*).

### NDGA

Nordihydroguaiaretic acid (NDGA), an ALOX15 pathway inhibitor, has been regarded to play an active role in STZ-induced DKD model. NDGA is more effective in improving renal function when the BG level is well controlled. Moreover, NDGA treatment can reduce the expression level of oxidative stress indexes in DKD (*Gad, 2012*).

**Table 1  Summary of ALOX15 inhibitors in a variety of diseases.**

| Inhibitors | Function | Diseases |
|---|---|---|
| Baicalin | ERK, ROS ↓ | Neurodegenerative disorder diseases |
|  | Behavioral deficits ↓ | Embolic stroke in rabbits; AIS |
|  | PPAR-γ expression ,translocation ↓ | Ischemia-reperfusion injury |
|  | VEGF ↓ | Angiogenesis of prostate cancer |
|  | Plasma 12-HETE ↓ | Vascular calcification |
| PD146176 | EMT ↓ | ECRS with nasal polyps |
|  | Inflammation, fibrosis, macrophages infiltration ↓ | UUO treatment of Alox15 knockout mice |
| BHPP | Urinary 12-HETE,12-HETE/cr excretion ↓ | DKD |
| NDGA | Oxidative stress index ↓ | DKD |
| ML351 | Glutamate oxidative toxicity ↓ | Ischemic stroke |
| ML355 | PAR-4 -induced aggregation, calcium mobilization, 12-HETE in β-cells ↓ | DM; Anti-platelet therapy |

### ML351

Quantitative high-throughput screening has identified ML351, a novel chemotype that inhibits ALOX15. Furthermore, kinetic experiments have shown that this class of inhibitors is a tightly bound hybrid inhibitor that does not diminish ferric ions in the active site. It has been found that ML351 protected HT22 cells from glutamate oxidative toxicity and mainly diminished cerebellar infarct size in the mice models of ischemic stroke (*Rai et al., 2010*).

### ML355

It has been reported that ML355 exhibited nM potency about ALOX15, indicating its outstanding selectivity against related LOX and COX. ML355 reduces 12-HETE expression level in mice/human β-cells mainly by inhibiting PAR-4-induced aggregation and calcium mobilization in the platelets of humans, suggesting a potential role for this inhibitor in animal models of DM and anti-platelet therapy (*Luci et al., 2010*).

The main functions of different ALOX15 inhibitors in various diseases are summarized in Table 1.

## Limitations and problems in using ALOX15 inhibitors
### Isomer specificity of ALOX15 inhibitors

First of all, most LOX inhibitors in Table 1 (such as NDGA, baicalin, PD146176) have no significant specificity, and some can inhibit a variety of LOX isomers. Therefore, it is impossible to back infer the biological activity of specific LOX isomers based on the results of inhibitor studies. Secondly, how to determine the specificity of LOX inhibitors is a problem. A recent study overexpressed each of the six rat Alox15 enzymes in HEK-293T cells and then tested a series of LOX inhibitors using cell lysates. The results showed that these LOX inhibitors (NDGA, baicalin, PD146176) exhibited only a low degree of isomeric specificity. However, some were previously considered isomer-specific inhibitors (*Gregus et al., 2013*).

### Homologous specificity of ALOX15 inhibitors

Since the discovery that mammalian ALOX isoforms play an important role in various biological effects, a large number of ALOX inhibitors have been developed. Unfortunately, most of them have neither parallel nor homologous features. Studies have shown that most ALOX inhibitors exhibit limited parallel specificity under rigorous and comparable experimental conditions. In addition, because of functional differences between mice and human ALOX homologs, inhibition of human ALOX homologs by an inhibitor does not necessarily mean that the corresponding mice homologs are also inhibited. For example, PD146176 was used as a specific inhibitor of ALOX15 in experimental strategies that completely failed to inhibit Alox15 in rats (*Gregus et al., 2013*). The reason may be related to the different location specificity of the two Alox15 homologs, but future experiments are needed to support this conclusion. Therefore, inhibitors that effectively interfere with human ALOX15 may not inhibit homologous enzymes in other species due to species specificity. For example, recent studies have shown that some oxazol-4-carbonitrile-based LOX inhibitors have high inhibition ability on ALOX15 in humans and mice, but can hardly inhibit other mammalian LOX subtypes (*Rai et al., 2014*). Thus, when interpreting the role of a given ALOX inhibitor in complex biological systems, consideration needs to be given to the lack of homologous specificity and varying degrees of homologous specificity of currently available ALOX inhibitors.

### Off-target effects of ALOX15 inhibitors

Some ALOX15 inhibitors (NDGA, baicalin) in Table 1 have antioxidant properties and may directly affect cellular redox homeostasis. The regulation of cellular redox state on gene expression has been discussed in genetic (*Brüne et al., 2013*) and epigenetic (*Goswami, 2013*; *Kim, Ryan & Archer, 2013*) level, so it is difficult to distinguish between the biological effects we observed due to inhibition of LOX or inhibition of the redox state. In conclusion, the results obtained with some ALOX15 inhibitors need to be interpreted to a certain extent and confirmed by further experiments of alternative functions.

## FUTURE QUESTIONS NEED TO BE DETECTED

To date, most of the literature on the role of ALOX15 in the occurrence of DM has focused on its expression and mechanism in glomerular mesangial cells, while its expression in other renal cells remains unclear. Therefore, it will be important to address the potential relationship between cell populations in the next step. Secondly, due to the dual role of ALOX15 in inflammation, is ALOX15 mainly an anti-inflammatory molecule or a pro-inflammatory molecule in the pathogenesis of DM? More researches are needed to determine whether the regulatory role of ALOX15 in DM is specific to DM or applies to other related metabolic diseases. Finally, we have to admit that trying to do something with ALOX15 is bound to create perturbations in other ways that are not considered. *In vivo* experiments demonstrate that Alox12 and Alox15 knockout mice are currently viable. Although the epididymal maturity of Alox15 defective sperm is irregular, the animals reproduce well and establish corresponding mouse colonies easily (*Walters et al., 2018*). On the other hand, the mild phenotype of Alox15, Alox12, and Alox5

knockout mice may be partly related to the fact that these two enzymes are not currently conditionally knocked out. One problem with unconditioned knockout mice is that they are produced to a certain extent selectively (only embryonic stem cells that survive the genetic manipulation are selected for blastocyst injection), and compensation mechanisms during early embryogenesis cannot be ruled out. In order to overcome these problems, induced knockout systems should be established, but such experiments are time-consuming and expensive, and need to be further explored in the future.

## RATIONALE AND PERSPECTIVE

ALOX15 is a vital enzyme involved in catalyzing fatty acid oxidation, which takes part in physiological and pathological procedures of the human body. In addition, ALOX15 can participate in the occurrence of DM and its complications through inflammatory reactions, oxidative stress, ferroptosis and other mechanisms, while the inhibition of ALOX15 expression can reduce the occurrence of DM and its complications. The specific mechanisms and treatment plans need to be more explored. In addition, studies have shown that ALOX15 inhibitors, such as baicalin, BHPP, NDGA have positive effects on different diseases *in vivo* and *in vitro* models. However, further clinical researches are needed to clarify the efficacy of these inhibitors in diseases. Therefore, in terms of scientific research, this study aims to provide research direction for researchers in DM-related research. In the clinical aspect, there is an urgent need to explore new and specific ALOX15 inhibitors, so as to provide a new direction for the diagnosis and treatment of DM and its complications.

## HIGHLIGHT

1. ALOX15 was initially found to be associated with the maturation of mitochondria in the later stages of erythropoiesis. Subsequent studies indicated that the expression of ALOX15 may also be involved in the occurrence of DM and its related complications.
2. ALOX15 can participate in the occurrence and development of various diseases through pro-inflammation or anti-inflammation, oxidative stress, ferroptosis and other functions.
3. ALOX15 can be used as a biomarker. Inhibition of the expression level of ALOX15 can help protect the kidney tissue and reduce the urinary protein level, which can be used as a new therapeutic target for the treatment of DM and its complications.
4. Baicalin, BHPP, NDGA and other ALOX15 inhibitors have positive effects on both *in vivo* and *in vitro* models of different diseases. Further studies are needed to develop and test specific pharmacological inhibitors of each LOX in order to apply them to therapeutic intervention of human diseases.

### Funding

This work was supported by the Lanzhou Science and Technology Bureau talent innovation and entrepreneurship (2021-RC-94); the National Natural Science Foundation of China (No. 81960142), the Youth Science and Technology Fund Program of Gansu Province (No. 21JR1RA157), the Talent Innovation and Entrepreneurship Project of Lanzhou City, Gansu Province (2021RCCX0027); the Lanzhou University Second Hospital Youth Fund (CY2021-QN-B01); and the Project of Department of Education of Gansu Province (2022B-050). Meanwhile, our experiments are supported by the Clinical Medical Research Center of Gansu Province (21JR7RA436). The funders had no role in study design, data collection and analysis, decision to publish, or preparation of the manuscript.

### Grant Disclosures

The following grant information was disclosed by the authors:
Lanzhou Science and Technology Bureau talent innovation and entrepreneurship: 2021-RC-94.
National Natural Science Foundation of China: 81960142.
Youth Science and Technology Fund Program of Gansu Province: 21JR1RA157.
Talent Innovation and Entrepreneurship Project of Lanzhou City, Gansu Province: 2021RCCX0027.
Lanzhou University Second Hospital Youth Fund: CY2021-QN-B01.
Project of Department of Education of Gansu Province: 2022B-050.
The Clinical Medical Research Center of Gansu Province: 21JR7RA436.

### Competing Interests

The authors declare there are no competing interests.

### Author Contributions

- Kaiying He conceived and designed the experiments, prepared figures and/or tables, authored or reviewed drafts of the article, and approved the final draft.
- Xiaochun Zhou analyzed the data, authored or reviewed drafts of the article, and approved the final draft.
- Hongxuan Du performed the experiments, prepared figures and/or tables, authored or reviewed drafts of the article, and approved the final draft.
- Jing Zhao analyzed the data, authored or reviewed drafts of the article, and approved the final draft.
- Rongrong Deng analyzed the data, prepared figures and/or tables, and approved the final draft.
- Jianqin Wang analyzed the data, authored or reviewed drafts of the article, and approved the final draft.

### Data Availability

This is a literature review and did not utilize raw data.

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
