# Peer review of "A review on the relationship between Arachidonic acid 15-Lipoxygenase (ALOX15) and diabetes mellitus"

_PeerJ, doi:10.7717/peerj.16239_

## Round 0.1 · original submission · Major Revisions

Please carefully revise the language style as it sounds quite repetitive at some points.

Reviewer 1 ·

Basic reporting

Arachidonic acid lipoxygenases (ALOX-isoforms) are lipid peroxidizing enzymes, which have been implicated in the pathogenesis of inflammatory, hyperproliferative and neurological diseases. In the human genome six functional ALOX genes (ALOX15, ALOX15B, ALOX12, ALOX12B, ALOXE3, ALOX5) have been detected and each of these genes encodes for a functionally distinct ALOX isoform. In mice a single orthologous gene exists for each human ALOX gene but in addition an Aloxe12 gene was detected in the mouse reference genome. In humans, this gene is a dysfunctional pseudogene. The present review is focused on one of these ALOX-isoforms (ALOX15), which has previously named 12/15-LOX, and its possible relation to the pathogenesis of diabetes mellitus. In principle, this topic is of direct medical relevance and writing a review on this issue is justified. The ms consists of four parts. 1. Brief introduction, 2. Biological roles of ALOX15, 3. ALOX15 and diabetes mellitus, 4. ALOX15 inhibitors. According to my personal judgement only part 3 (ALOX15 and diabetes) is of good quality whereas the other parts require extensive rewriting. These parts are written in a way that indicates that the authors of this review paper not well experienced in the field of lipoxygenase research. However, they did a good job in putting together the most relevant original reports on the role of ALOX15 pathway in the pathogenesis of diabetes.

Major points of criticisms
1. ALOX nomenclature: The authors employ arachidonic acid based ALOX nomenclature. According to the recommendations of the nomenclature committee of the International Union of Biochemistry proteins should be named according to their encoding genes, when the genes encode for a single functional protein. This is the case for all mouse and human ALOX isoforms and thus, 12/15-LOX should be named ALOX15. This protein is encoded by the ALOX15 gene. In addition to ALOX15 five further ALOX-isoforms are present in humans: ALOX15B (previously named 15-LOX-2), ALOX12 (previously named platelet-12-LOX), ALOX12B (previously named 12R-LOX), ALOXE3 (previously named epidermal LOX3), ALOX5 (previously named 5-LOX). The previously used arachidonic acid-based lipoxygenase nomenclature is outdated and should not be used any more since it stresses a single catalytic property of the enzymes but does not consider the evolutionary relatedness between the different ALOX isoforms.

2. Differences between mouse and human ALOX-isoforms
Human ALOX15 (12/15-LOX) exhibits a dual reaction specificity with arachidonic acid forming 15-HpETE as major oxygenation product. In contrast, corresponding mouse ortholog oxygenates this substrate mainly to 12-HpETE. Thus, the product profile of mouse and human ALOX15 is quite different. In fact, most (>95 %) mammalian ALOX15 orthologs are arachidonic acid 12-lipoxygenating enzymes and only highly developed Hominidae (orangutans, gorillas, chimpanzee, H. sapiens, H. denisovans, H. neanderthalensis) express arachidonic acid 15-lipoxygenating ALOX15 orthologs. This functional difference between human and mouse ALOX must be considered when experimental data obtained in mouse models of human diseases are transferred into the human situation. Unfortunately, the authors did not always consider this functional difference when they interpret the results obtained in mouse or rat disease models

3. ALOX15-GPx4 interaction
In chapter 2.2. the authors discuss the functional relation of ALOX15 and GPx4. Unfortunately, the authors did not reference a singly paper indicating the direct functional interplay of the two enzymes. How exactly does GPx4 control the catalytic activity of ALOX15? Is there a special mechanism of protein-protein interaction.

4. ALOX15 inhibitors
The authors discuss a number of different ALOX15 inhibitors. Unfortunately, the isoform- and ortholog-specificity of these inhibitor has not been critically reviewed. Moreover, the authors did neither discuss the potential off-target effects of the different compounds (see Biomed Pharmacother. 2022 Jan; 145:112434. doi: 10.1016/j.biopha.2021.112434). A more critical evaluation of the usefulness of the currently available ALOX15 inhibitor would be helpful for readers who want to employ such compounds as mechanistic tools for future studies.

In addition to these major points of criticisms I have a number of more specific issues that should be addressed by the authors in case the handling editor invites resubmission of a suitably revised version of this ms.

a) Line 82: … non-heme ferritin: This phrase is misleading. Ferritin is an iron storage protein that involves up to 5000 iron ions per apoferritin molecule. Lipoxygenases contain a single iron ion per enzyme molecule, which is of catalytic relevance. Ferritin occurs in cells but also in the blood plasma but lipoxygenase exclusively occur inside of cells. The iron binding in ferritin and in lipoxygenases are very different and thus, lipoxygenase should not be named “kind of non-heme ferritin”.

b) Line 88: The arachidonic acid based ALOX nomenclature (5-LOX, 8-LOX, 12-LOX, 15-LOX) is outdated and should not be used any more (see comment 1). In humans 5-LOX is ALOX5, 8-LOX is ALOX15B, 12-LOX is either ALOX12 or ALOX12B, 15-LOX is ALOX15. I strongly encourage the authors to use the gene based ALOX nomenclature.

c) Line 89: “Amino acid sequence homology of 12-LOX and 15-LOX reached 80-90 %” This statement is wrong or at least misleading. Human ALOX15 and mouse Alox15 share a 86 % amino acid homology and similar degrees of amino acid conservation was found for the ALOX15 orthologs of rats, pigs and cattle: Appropriate references for these enzymes should be given. Despite this high degree of amino acid sequence homology mouse and human ALOX15 orthologs have different functional properties (15-lipoxygenating vs. 12-lipoxygenating). On the other hand, human ALOX15 and human ALOX12 share a much lower degree (65%) of amino acid sequence homology. These data clearly indicate that mammalian ALOX15 orthologs are much closer related to each other than human ALOX15 and human ALOX12.

d) Line 90: “There are three types of 12/15-LOX.” This statement is also wrong. The human platelet type 12-LOX is ALOX12 and has nothing to do with human 12/15-LOX. Both enzymes are products of different genes and exhibit very distinct catalytic properties. There is no leukocyte-type 12/15-LOX in humans and the human equivalent of the mouse, rat and pig leukocyte-type 12/15-LOX is ALOX15. In the epidermis of humans 3 different ALOX-isoforms are expressed (ALOX15B, ALOX12, ALOXE3). In mice the orthologous isoforms are found in the skin (Alox12, Alox15b which is sometimes also called 8-LOX and Aloxe3) but in this species Aloxe12 is also present, which is absent in humans.

e) Line 94: 75 KB should be 75 kbp.

f) Line 104: Lipoxins, hepoxilins and eoxins are not bioactive proteins they are bioactive lipids.

g) Line 108: To the best of my knowledge in ref 9 simultaneous incubation of human ALOX15 with an equimolar mixture of linoleic acid and arachidonic acid was not carried out. In fact, unpublished data from our lab suggest that an equimolar mixture of these two substrates is converted by recombinant human ALOX15 mainly to 13-HpODE. In fact, 15-HpETE is only a minor side product. These data suggest that linoleic acid is a better substrate for this enzyme than arachidonic acid. This result contrasts the conclusion drawn in this ms.

h) Line 110: The dominant linoleic oxygenation product of human ALOX15 (12/15-LOX) is 13-HpODE. 9-HODE is only produced in trace amounts (Fig. 2D in ref. 9).

i) Line 121: Lipoxygenase cannot produce arachidonic acid. They can oxygenate this fatty acid (use it as substrate) but they cannot synthesize it.

j) Line 122: Lipoxins are not proteins but lipids and thus, they cannot be expressed. They are synthesized from arachidonic acid via different ALOX pathways.

k) Line 131: The biosynthesis and the biological relevance of lipoxins has recently been challenged (Front Pharmacol. 2022 Mar 2;13:838782. doi: 10.3389/fphar.2022.838782) and for the sake of fairness this reference should be included into the reference list

l) Line 134: 9-HPODE is not a major ALOX15 product (see comment h).

m) Line 151: GSH-peroxidases (GSH-Px) are peroxide reducing enzymes (not peroxidase decomposing enzymes).

n) Line 152: The biological role of GSH-Px is to reduce organic and inorganic peroxides to the corresponding alcohols using GSH as electron donor oxidizing this reductant to GSSG. This sentence needs to be rephrased.

o) Line 171: Ferroptosis. This chapter gives the impression that ALOX15 is an essential player in the mechanism of ferroptosis. This is, however, not the case. In a large number of cellular model systems there is clearly ferroptosis although ALOX15 is not expressed in these cells. However, in cellular systems expressing this enzyme ALOX15, may play a role for ferroptotic cell death.

p) Line 225: The authors report that in human islet cells an increase in exogenous 12S-HEPE and 12S-HETE leads to a decline in insulin production. This might be the case but since human ALOX15 produces mainly 15S-HPETE from arachidonic acid it remains unclear whether this observation is of any biological relevance. 12S-HPETE and 12S-HETE are only minor side products (10%) of human ALOX15 pathway. Why should the minor side product be of higher biological relevance than the main product (15-HPETE/15S-HETE) of this pathway?

q) Line 237: 5S-HETE is not an ALOX15 product. It is dominantly formed by ALOX5.

r) Line 242: … diverse phases (not phrases) of DR.

s) Line 352: In the highlight section but also earlier on in the ms the physiological role of ALOX15 in cell differentiation and maturation (erythropoiesis) should at least be mentioned.

t) Are ALOX15 orthologs expressed in all mammals. We know that corresponding genes occur in rabbits, humans, mice, rats, pigs and other mammalian species but are there mammals lacking ALOX15 genes. If this is the case these mammals should be protected from diabetes mellitus.

u) The reference list should be updated. There is no mentioning of the pioneer work of Rapoport and Schewe (Berlin), who first described a mammalian ALOX15 ortholog (rabbit ALOX15) back in 1975. They purified this enzyme from rabbit reticulocytes (1979) and implicated it in erythropoiesis. Independent experiments with ALOX15 knockout mice recently supported these conclusions. The ms also lacks any refence to the work of the E. Sigal group (San Francisco), who first characterized mouse and human ALOX15 and explored the molecular basis for the different reaction specificity of mouse and human ALOX15 orthologs.

In the light of these comments, I feel myself unable to recommend publication of this ms. However, if the authors can address the critical points raised in this evaluation report a suitably revised version of the ms should undergo a second round of evaluation.

Experimental design

The study design is OK but the reference list lacks important papers in the field.

Validity of the findings

This is a review article that does not provide new findings.

Additional comments

see attached evaluation report (section1)

·

Basic reporting

In this literature review the authors tried to summarize the role of 12-15LOX in the pathobiology of DM and its complications.
The biochemical part is fine.
Language is ok.
More literature survey looking at the effect of individual products of 12-15LOX pathway is needed.

Experimental design

Ok and reasonable but still lacks depth of the survey.

For instance the authros looked only at 12-15-LOX products. But it need to be noted that cells/tissues/organs do not work in isolation and so a comprehensive look at AA metabolism in DM is needed.
Do the authors think that when pro-inflammatory metabolites are produced the cells/tissue will not produce anti-inflammatory metabolites to counteract the same.
Cells and tissues will always try to maintain a balance between pro and anti-inflammatory metabolites and limit tissue damage, So such an approach and argument needs to be developed.
If LXA4 is produce din excess what happens and if excess of hepoxilins produced what happens. How the cells and tissue determine which molecule to be produced and why? what are the circumstances when one type of molecule(s) are produced in excess and why he counter measures fail.
What happens when AA excess is there or AA deficiency is there.
Why AA excess or deficiency occurs and how it is mitigated by the cells and tissues.
Some of these aspects need to be discussed.

Validity of the findings

Valid but ened more depth as detailed above.

Additional comments

THe authors did not discuss recent studies that AA and LXA4 may have anti-diabetic actions.

---

## Round 0.2 · Minor Revisions

Thanks for the work you have done to improve this manuscript. As you can see by reviewers' comments, I would kindly ask you to work additionally on some modifications. Please consider to clearly state the limitations when discussing your ideas, and clearly present as hypothesis and future perspective what is not supported by data.

Reviewer 1 ·

Basic reporting

The review is of cross-disciplinary interest and in the scope of the journal.

this field has not been reviewed recently.

Introduction is OK.

The authors have adequately addressed my critical comments and modified the ms accordingly. However, there are still some issues to be considered. I am appending a commented pdf of the revised ms, in which the required altereations are clearly marked. After these alterations have been made the ms can be published without further review.

Experimental design

Survey methodology is OK.

The updated reference list is OK!

The review is well organized.

Validity of the findings

This is an interesting review that summarizes the current knowledge in the field.

The review asks unsolved questions and suggests further directions of research.

Annotated reviews are not available for download in order to protect the identity of reviewers who chose to remain anonymous.

·

Basic reporting

I read the revised manuscript.
In the absence of clear cut experimental evidence of the effect of various metabolites of AA in the occurrence of DM and it’s complications, the review remains conjectural.
There are no studies that measured all the metabolites of AA in those with DM not studies are available that looked at the effects of various metabolites of AA on the incidence of DM. Such data is needed to talk about the role of AA metabolites especially if 15-lox in DM and it’s complications.
The authors talk s as out only 15-lox metabolites but do no take into consideration the other products such as 5-lox and Cox metabolites.’
It is important to note that there are always dynamic interactions and equilibrium among various metabolites of not only AA but also of EPA, DHA, LA, etc. so emphasizing on just one pathway is not a good idea.
If some manipulation of the 15-lox is attempted there is bound to be perturbations in other pathways that is no taken into consideration.
I suggest that others abridge their review by more than 50% and discuss the above mentioned limitations and make it a brief hypothesis and also discuss how their ideas can be tested snd what results are expected of such experiments.

Experimental design

See my comments above.

Validity of the findings

Not sufficient

Additional comments

See above.
More balanced view and discussion is needed.

---

## Round 0.3 · Minor Revisions

I acknowledge the changes made based on the reviewers' suggestions. Before acceptance, I kindly ask the authors to carefully edit the manuscript for language, repetitions, use of abbreviations, spaces before and after punctuation, and brackets to make the text clear and readable. Please find attached a file with some corrections as an example.

**Language Note:** The Academic Editor has identified that the English language must be improved. PeerJ can provide language editing services - please contact us at copyediting@peerj.com for pricing (be sure to provide your manuscript number and title). Alternatively, you should make your own arrangements to improve the language quality and provide details in your response letter. – PeerJ Staff

---

## Round 0.4 · Minor Revisions

Thanks for your work on the text. My final suggestion is to move the last part of the introduction to a different section. For instance, apologizing to authors that cannot be mentioned is not appropriate in a scientific paper. Instead, clearly define in the methodology search your inclusion and exclusion criteria for the papers considered. The sentence "We will do our best to be neutral in this article" is not appropriate as it implies you were not neutral in your paper selection and/or considerations. If you have some conflicts to declare, please make them explicit in the dedicated section. Lastly, move the explanation about the gene nomenclature to the methods section (and maybe give a short explanation of why you chose to do so).

---

## Round 0.5 · accepted · Accept

I thank the authors for their effort in implementing the manuscript. The text can now be accepted for publication.